# Preclinical Models of Low-Grade Gliomas

**DOI:** 10.3390/cancers15030596

**Published:** 2023-01-18

**Authors:** Pushan Dasgupta, Veerakumar Balasubramanyian, John F. de Groot, Nazanin K. Majd

**Affiliations:** 1Department of Neurology, Dell Medical School, University of Texas at Austin, Austin, TX 78712, USA; 2Department of Neuro-Oncology, UT MD Anderson Cancer Center, Houston, TX 77030, USA; 3Department of Neurosurgery, University of California San Francisco, San Francisco, CA 94143, USA

**Keywords:** glioma, low-grade glioma, preclinical models, IDH-mutant glioma, patient avatars

## Abstract

**Simple Summary:**

Preclinical models are essential for the advancement of our understanding of glioma biology and the development of novel therapeutics. Much of our progress in the treatment of low-grade glioma has been hampered by our limited ability to develop ideal preclinical models. This has proven to be a formidable task given the complex factors one must account for, such as genetic background, intratumoral heterogeneity, intact blood–brain barrier, and the tumor microenvironment. As new knowledge is acquired regarding low-grade glioma, preclinical models must be refined and adjusted to reflect the actual biology of human glioma as closely as possible. In this review, we delve into in vitro and in vivo models of low-grade glioma with particular attention to illuminating the multifaceted task of developing the most optimal models.

**Abstract:**

Diffuse infiltrating low-grade glioma (LGG) is classified as WHO grade 2 astrocytoma with isocitrate dehydrogenase (IDH) mutation and oligodendroglioma with IDH1 mutation and 1p/19q codeletion. Despite their better prognosis compared with glioblastoma, LGGs invariably recur, leading to disability and premature death. There is an unmet need to discover new therapeutics for LGG, which necessitates preclinical models that closely resemble the human disease. Basic scientific efforts in the field of neuro-oncology are mostly focused on high-grade glioma, due to the ease of maintaining rapidly growing cell cultures and highly reproducible murine tumors. Development of preclinical models of LGG, on the other hand, has been difficult due to the slow-growing nature of these tumors as well as challenges involved in recapitulating the widespread genomic and epigenomic effects of IDH mutation. The most recent WHO classification of CNS tumors emphasizes the importance of the role of IDH mutation in the classification of gliomas, yet there are relatively few IDH-mutant preclinical models available. Here, we review the in vitro and in vivo preclinical models of LGG and discuss the mechanistic challenges involved in generating such models and potential strategies to overcome these hurdles.

## 1. Introduction

Diffuse gliomas are the most common malignant tumors of the central nervous system (CNS) in adults [1]. They make up about 30% of all brain and 80% of all malignant brain tumors [2,3]. Diffuse low-grade gliomas account for approximately 15% of all gliomas in the United States and have a 5-year survival rate ranging from 30% to 80% [1]. Glioblastoma (WHO grade IV) is the most common adult glioma, accounting for 15% of all primary brain and central nervous system tumors and has a 2-year survival rate of 26.5% [1,4].

Diffuse gliomas are heterogenous groups of tumors that are universally incurable despite multimodality standard-of-care treatments, which include surgery, radiation therapy, and chemotherapy. There is an unmet need to develop novel therapeutics to improve patient outcomes. To achieve this goal, it is imperative to improve our understanding of glioma biology and to establish preclinical models that resemble the clinical and molecular characteristics of human glioma as closely as possible. Creating effective models for low-grade glioma has revealed unique challenges, as these tumors are slow growing, unlike their higher-grade counterpart, glioblastoma. Many researchers have contributed to our current ability to create effective models; however, there remains much more to do.

In this review, we will discuss the grading, classification, and molecular pathology of low-grade gliomas, the knowledge of which is important in understanding preclinical models of gliomas. We also explain properties of an ideal preclinical model and existing in vitro and in vivo models of low-grade gliomas.

### 1.1. Diffuse Low-Grade Gliomas

In the WHO 2021 classification, there are three primary categories of adult-type diffuse gliomas: isocitrate dehydrogenase (*IDH*)-mutant, 1p/19q codeleted oligodendroglioma; IDH-mutant, non-codeleted astrocytoma; and IDH-wildtype glioblastoma [5]. Diffuse low-grade glioma is classified as WHO grade 1 and 2 astrocytoma with *IDH* mutation or oligodendroglioma with *IDH* mutation and 1p/19q codeletion.

### 1.2. Grading and Classification of Gliomas

The grading and classification of gliomas have gone through several changes throughout the years. The first grading system was developed by Albert Broders at the Mayo Clinic and employed a numerical grading system dividing tumors into four histological grades of malignancy [6]. This system did not take into account clinical history. The histological grading system uses the “AMEN” score, which consists of nuclear atypia (A), mitosis (M), microvascular/endothelial proliferation (E), and necrosis (N) to evaluate malignancy [7]. In this system, grade 3 tumors required a significant mitotic count, while grade 4 tumors required microvascular proliferation or necrosis. However, this histological grading system was intrinsically subjective, with inter- and intra-observer variabilities [8]. The WHO grading system strived to provide a biology-oriented grade which was based more on estimated clinical outcome. The 2016 WHO classification of CNS tumors introduced integrated molecular and histological diagnoses to classify CNS tumors, and the 2021 WHO classification further expanded upon molecular features [9].

### 1.3. Molecular Pathology of Low-Grade Gliomas

The presence of the *IDH1* mutation has become a defining factor for adult diffuse low-grade glioma. Roughly 70% of grade 2–3 gliomas harbor mutations in either *IDH1* or its mitochondrial counterpart *IDH2* [10]. Often, these tumors are seen in younger patients and carry a better prognosis [11].

The normal function of IDH is to convert isocitrate to alpha-ketoglutarate (α-KG) [12]. Mutations in *IDH1* or *IDH2* are exclusively missense mutations of the arginine residues in the active site of the enzyme, which is R132 for IDH1 and R172 for IDH2 [10,13]. These mutations lead to neomorphic enzymatic activity, resulting in the production of the oncometabolite D-2-hydroxyglutarate (2-HG) from alpha-ketoglutarate (α-KG) [12]. The oncometabolite 2-HG inhibits a large number of α-KG-dependent enzymes involved in fatty acid synthesis and maintenance of redox potential and results in metabolic stress in IDH-mutant tumors [14,15]. In addition, 2-HG also inhibits DNA and histone demethylation, which results in a hypermethylated epigenetic state (G-CIMP phenotype) leading to impaired cell differentiation and dysregulation of oncogenes and tumor suppressor genes [16,17,18].

### 1.4. IDH-Mutant Astrocytoma

IDH-mutant astrocytoma encompasses grade 2 and grade 3 tumors that have *IDH1* or *IDH2* mutations without 1p/19q codeletion. Morphologically, these tumors are hypercellular and composed of diffusely infiltrative fibrillary glial cells [9,19]. The WHO does not provide a firm definition for a significant mitotic rate that would characterize a grade 3 tumor. However, in general grade 3 tumors will have a higher mitotic rate than grade 2 tumors and also display histologic features of anaplasia. Grade 4 IDH-mutant astrocytomas are defined by the presence of microvascular proliferation, necrosis, and/or homozygous deletion of cyclin-dependent kinase inhibitor 2A/B (*CDKN2A/B*) in IDH-mutant astrocytoma.

Beyond *IDH1* and *IDH2* mutations, adult-type *IDH*-mutant low-grade astrocytomas commonly harbor inactivating mutations in *TP53* and *ATRX*. *TP53* is the most frequently mutated gene in cancer. Its product, p53, is a tumor suppressor with roles as a regulator of cell cycle arrest and apoptosis [20,21]. ATRX is a regulator of chromatin remodeling and transcription and is known to form a chromatin remodeling complex with death domain-associated protein (DAXX), leading to the deposition of H3.3 in telomeric regions, pericentric heterochromatin, and various regions of repeat DNA [22,23]. One explanation for the co-occurrence of *TP53* and *ATRX* mutations is that cells deficient in ATRX undergo p53-dependent cell death [24,25]. ATRX-deficient tumors exhibit a pathological form of telomere maintenance whereby telomeres are lengthened in a telomerase-independent process called alternative lengthening of telomeres [26]. Intriguingly, there are rare infratentorial variants of IDH-mutant astrocytoma that have distinct molecular and clinical characteristics with relatively worse prognosis [9].

### 1.5. IDH-Mutant Oligodendroglioma

*IDH*-mutant oligodendroglioma is defined by the presence of 1p19q codeletion in the context of either *IDH1* or *IDH2* mutation and are either grade 2 or 3. Compared with grade 2 tumors, grade 3 tumors have histological features such as increased cellularity, marked atypia, greater mitotic activity, microvascular proliferation, and necrosis [27]. Genetically, the presence of homozygous *CDKN2A* and/or *CDKN2B* deletion classifies an oligodendroglioma as grade 3 [28]. Mutations in the TERT promoter occur frequently in these tumors. The catalytic subunit of telomerase is encoded by *TERT*, and point mutations in the promoter lead to TERT overexpression and thus telomere elongation [29,30]. Abnormal telomere maintenance seems to be a central process in gliomagenesis, which is implied by the mutual exclusivity of *TERT* promoter and *ATRX* mutations in IDH-mutant adult gliomas. *CIC* gene mutations are seen in up to 70% of oligodendrogliomas and have been linked to worse survival [31,32]. Such mutations interrupt the CIC protein’s repressor functions by rendering the protein truncated, degraded, or non-functional [32]. This leads to upregulation of the ETS-Pea3 family of transcription factors, ETV1, ETV4, and ETV5, which are known oncoproteins and have been shown to induce cell proliferation in melanoma, prostate cancer, and gastrointestinal stromal tumors [33,34,35]. Transcriptomic analysis of CIC-mutant gliomas demonstrated an upregulation of the ETS/Pea3 family of proteins, which are normally repressed by the wildtype CIC protein [32,36]. Together, these findings suggest that the aggressiveness of CIC-mutant subsets of oligodendroglioma may be due to the inactivation of the repressor effects of CIC and increased expression of the ETS family of oncoproteins, which are involved in cell proliferation and invasion.

Inactivating *FUBP1* mutations also occur in 15% to 30% of oligodendrogliomas [31]. FUBP1 loss has been shown to cause widespread changes in both RNA splicing and expression of aberrant driver isoforms [37], but its precise role in gliomagenesis is not clear. Oligodendrogliomas with both *CIC* and *FUBP1* mutations are exceedingly rare. Further studies are needed to establish the relationship, if any, between these inactivating mutations and chemosensitivity in these tumors. Understanding the molecular genetics of IDH-mutant gliomas and their association with prognostic risk stratification is crucial in our interpretation of data generated from preclinical models.

### 1.6. Ideal Preclinical Model of Low-Grade Gliomas

In order to make new biologic discoveries of low-grade gliomas and to make progress in developing novel therapeutics, it is imperative to have preclinical models that accurately recapitulate the human disease. Beyond the fundamentals of reproducibility and stability, an ideal preclinical model should demonstrate genetic background, intratumoral heterogeneity, and a microenvironment that closely resemble those of human diffuse glioma. However, developing an ideal model that captures all these elements in vivo has been challenging. In vitro generation of murine IDH-mutant cell lines or creating stable patient-derived IDH-mutant lines that retain the IDH-mutant status have proven to be elusive.

## 2. Cell Culture Models

### 2.1. Murine Cell Lines

An early model of diffuse glioma from the 1970s relied on carcinogen-induced gliomagenesis, where N-ethyl-nitrosourea (ENU) was injected into pregnant rats [38]. It is thought that the in utero exposure of embryos to ENU induces brain tumors, as injection of ENU in adult animals does not lead to the generation of brain tumors [39]. There are several key mutations that drive the ENU-induced glioma formation in rats, including *Braf*, *Tp53*, and *Pdgfrα* mutations; *Cdkn2a* deletion; and *Egfr* amplification [40]. Even though these molecular aberrations are also commonly seen in human diffuse gliomas, the ENU-induced model of gliomagenesis has poorly reproducible characteristics of glioma formation [41]. Several researchers took advantage of the heterogeneity derived from the use of alkylating agents to generate diverse glioma models. This led to the creation of many different murine glioma cell lines. Among the more commonly used are C6 glioma, 9L gliosarcoma, T9 glioma, RG2 glioma, F98 glioma, CNS-1 glioma, and BT4C glioma [42]. These lines do not harbor IDH mutations and are considered high-grade glioma models, which demonstrates that this method is not suitable for generating IDH-mutant murine cell lines.

Another method used to generate glioma murine lines is the CRE/LOX system. Bardella et al. created a murine glioma model that had conditional, inducible expression of the mutant Idh1 in the subventricular zone (SVZ) stem cell niche in the adult mouse. Since previous researchers had noticed that mutant *Idh1* knock-in mice under the control of nestin died perinatally and exhibited brain hemorrhages [43], the researchers created a tamoxifen-inducible system and initiated the tamoxifen at 5 to 6 weeks of age [44]. In this manner, the mice survived, and the resulting cells were more proliferative and displayed invasive characteristics. However, attempts to culture IDH1-mutant neurospheres from the SVZ of these mice were not successful. The authors therefore grew primary SVZ cells dissected from IDH-wildtype mouse pups and stably expressed mutant IDH1.

### 2.2. Patient-Derived Cell Lines

Patient-derived cell lines are a common tool used across cancers including glioblastoma, which are commonly grown in culture as tumor spheres. There has been a great need for human patient-derived cell lines that endogenously express mutant *IDH1* with the ability to initiate tumors in mice that also retain low-grade genetic characteristics. Far fewer *IDH*-mutant than *IDH*-wildtype lines are available, as it has been more challenging to develop such lines. Luchman et al. took a resected tumor from a patient with grade 3 IDH-mutant astrocytoma and generated a stable cell line that retained the IDH1 mutation and exhibited self-renewal and multipotency [45]. The authors named this neurosphere line BT142. These cells were injected into the striata of NOD/SCID mice, which formed tumors with cells that were poorly differentiated with enlarged hyperchromatic irregularly round to angulated nuclei and scant cytoplasm. The researchers found that growth was much faster in vivo than in vitro. In fact, they performed serial xenografts using cells from the xenografted tumor and were able to propagate the line while retaining the mutant IDH1 status. The 2-HG to 2-KG ratio in conditioned medium of this line demonstrated that this model recapitulated metabolic alterations expected in IDH-mutant cells. Another notable IDH-mutant line is TS603, which was generated from a patient with grade 3 IDH-mutant oligodendroglioma harboring a codeletion of 1p and 19q [46].

## 3. Murine Models

### 3.1. Murine-Derived Genetically Engineered Mouse Models

Our growing knowledge of driver mutations involved in gliomagenesis has led to the development of genetically engineered mouse models and murine cell lines from them. One prominent method for developing these genetically engineered mouse models uses viral vectors to initiate tumor formation through the delivery of cancer-initiating genes. This system allows for glia-specific gene transfer in vivo using replication-competent ALV splice acceptor (RCAS) viral vectors and a transgenic mouse line (Gtv-a) that produces the receptor for ALV-A (TVA) from the astrocyte-specific promoter for the gene encoding glial fibrillary acidic protein (*Gfap*) [47]. Philip et al. used an RCAS/TVA mouse glioma model to demonstrate in vivo that *IDH* R132H promotes gliomagenesis [48]. However, Idh1 mutation alone was not sufficient to drive tumor development. Glioma development did not occur when mutant Idh1 was expressed in a genetic background with loss of *Cdkn2a*, *Atrx*, and *Pten* in vivo. However, the addition of PDGFA expression in this combination resulted in glioma development in 88% of injected mice. When wildtype Idh1 was expressed instead of mutant Idh1 in the same background, only 20% of the injected mice developed glioma. These data support a context-specific role of mutant IDH1 as a promoter of gliomagenesis when it is able to work with PDGFA in a genetic background of *Cdkn2a*, *Atrx*, and *Pten* loss.

### 3.2. Patient-Derived Xenograft Models

Another approach for developing murine models of low-grade glioma is to use tissue from patients. In this xenograft approach, the idea is to take glioma cells from patient tumors and then grow them in mice. In order to create a more context-specific model, orthotopic xenograft systems are created where patient-derived tumor cells are grown in the brains of mice. Klink et al. took tumor tissue from a patient with recurrent grade 3 oligodendroglioma, enzymatically dissociated it, and then grew cell aggregates in serum-free medium with epidermal growth factor and fibroblast growth factor [49]. They subsequently injected these cells intracranially into eGFP NOD/SCID mice. Tumors that grew in the mice demonstrated a diffusely infiltrating tumor pathology with round tumor nuclei and clear cytoplasm, which were consistent with oligodendroglioma histology. Genetic comparisons between the patient’s tumor and the xenograft tumor showed maintenance of the 1p and 19q losses in addition to maintenance of losses on chromosomes 6, 11, and 14. Exome sequencing of the patient tumor and the xenograft revealed that they both had mutations in *IDH*, *FUBP1*, and *CIC*. However, the xenograft did not perfectly match the patient’s tumor, as there were gains on chromosomes 11 and 4q. This work represented the first model in which a human oligodendroglioma could maintain its histological and molecular features in an intracranial mouse xenograft over serial passages.

Zeng et al. looked at the differences in generating patient-derived xenograft models in mice using tissue from patients with different grades of glioma. They took tumor tissues from 16 patient tumors of various grades and implanted them in mice, from which they established 11 glioma xenograft models. Not surprisingly, the researchers found that higher grades were associated with greater success in generating xenografts, with success rates of 33.33% for grade 2, 60.0% for grade 3, and 87.50% for grade 4 [50]. IDH-wildtype status and high Ki67 expression correlated with greater success in generating xenografts. These xenografts recapitulated the major histologic and molecular characteristics and key immunophenotypic features of the original tumors.

Navis et al. generated a mutant IDH1 oligodendroglioma xenograft line, which they characterized from a histological and metabolic standpoint. Though this line was established before the role of mutant IDH1 was discovered, the xenograft tumors were found to produce elevated levels of the oncometabolite D-2HG [51], as would be expected in an IDH-mutant line.

## 4. Emerging Models

### 4.1. Patient Avatars of Low-Grade Gliomas

An alternative model to cancer cell lines or mouse models is organoid models. They consist of tissue spheroids derived from progenitor cells or processed from tumor resections [52,53]. Unlike relatively homogeneous 3D spheroids, organoids are composed of multiple cell types, allow for self-organization, differentiation, mixed heterogeneity, and recapitulate features of in vivo cell growth all within the culture environment [54]. Thus, organoid models have the attractive combination of both supporting higher throughput studies and also maintaining diverse cell populations [55]. Recently, Abdullah et al. established a patient-derived LGG organoid model [55]. They collected specimens from brain tumor resections in 15 LGG patients which were then parcellated and placed in specialized media and cultured under 5% oxygen for 4-24 weeks before processing and analysis. The LGG organoid model recapitulated the histological features, molecular markers of stemness, proliferation, and vascular composition of the primary tumor. Moreover, they were also capable of maintaining parental tumor cellular heterogeneity, proliferative capacity, and distinctive genomic alterations [55]. Thus, patient-derived LGG organoids represent a unique patient avatar of LGG.

### 4.2. The Promise of In Silico Models

Despite the best preclinical biological models, the issues of cost, human effort, and time all limit the ease of advancements. In silico approaches offer another means of modeling LGG that circumvent these issues. There is tremendous promise in this area that some studies are beginning to unearth. A proof of concept has been shown in a recent study where a mathematical model of LGG response to temozolomide (TMZ) and radiation therapy (RT) was constructed to carry out in silico experiments to explore different treatment regimens [56]. They used longitudinal imaging data from LGG patients to obtain patient-specific parameters. Using their models, computer simulations showed that concurrent cycles of TMZ and RT could provide better therapeutic efficacy than concomitant radio-chemotherapy [56]. The authors found clinical trial validation of their in silico results in the clinical trial by Van den Bent et al., where it was found that deferring RT in LGGs did not alter survival time [56,57]. This study provides evidence that in silico models can be useful in the study of low-grade gliomas. It also underscores the potential that these in silico preclinical models can provide.

## 5. Challenges and Strategies in the Development of Low-Grade Glioma Models

Developing models of low-grade glioma has been a challenging endeavor. One major challenge has been in modeling the appropriate genetic background. Different types of vectors for gene delivery have been utilized by different groups to accomplish this (Table 1). Another challenge has been in the incorporation of the immune microenvironment. Various strategies have been employed to address these difficulties resulting in multiple LGG models (Table 2).

### 5.1. IDH Status

In order to study the role of IDH in low-grade glioma in vivo, Sasaki et al. generated *Nes-Idh1^R132H/wt^* (*Nes-KI*) mice and control *Nes-Idh1^wt/wt^* (*Nes-WT*) mice using Cre Lox technology with the *Nestin* promoter as the driver for expression of mutant IDH [43]. Their work was the first in vivo study to demonstrate that the metabolite D2HG, which is associated with the *Idh1* R132H mutation that inhibits mouse embryonic brain development. Notably, brain-specific expression of this *Idh1* mutation caused brain hemorrhage and perinatal death. Intracellular ROS levels were dramatically reduced in *Idh1-KI* brain cells, which also had an elevated NADP^+^/NADPH ratio and catalase activity. D2HG was found to impair collagen maturation, disrupting basement membrane structure and prompting a stress response in the endoplasmic reticulum. The authors concluded that D2HG associated with the Idh1-mutant enzyme may function as an oncometabolite that induces HIF target gene transactivation, disrupts collagen maturation, and impairs basement membrane structure [43].

The same researchers wanted to develop an IDH-mutant glioma mouse model. Since the *Idh1^R132H/wt^ (Nes-KI)* line was lethal, they crossed *Idh1^LSL/wt^* mice with *GFAP-Cre* mice to generate *GFAP-KI* mutants. In contrast to *Nes-KI* mice, which could not survive to adulthood, some of the *GFAP-KI* mice did. This created an opportunity to establish a model for low-grade IDH R132-mutant glioma. However, the GFAP-KI mice had a much shorter lifespan than their controls and did not develop glioma. The authors interpreted this to indicate that either mutated IDH alone is insufficient for gliomagenesis or that the mice did not live long enough to develop gliomas. The authors attempted to cross GFAP-KI mice with mice harboring deletion of the tumor suppressor *Trp53* to enhance glioma formation, but these mice displayed a broad spectrum of systemic tumors due to the leaky expression of CRE. This demonstrated the challenge of generating *Idh1*-KI mice that express mutant IDH1 protein exclusively in the brain and can be crossed with tumor-prone mice.

To circumvent the issue of embryonal lethality, Pirozzi et al. generated a mutant *Idh1* conditional knock-in model that produced mice heterozygous for the mutant *Idh1* [63]. This was conducted using a targeting vector containing a stop cassette flanked by LoxP sites and the *Idh1*- 132H mutation. The preceding LoxP-flanked stop sequence blocks the expression of the modified allele, resulting in a knocked-out allele that is restored through Cre-recombinase-mediated excision of the stop cassette. Neural stem cells (NSCs), which are found in the SVZ of the lateral ventricles, are a purported cell of origin for glioma and are known to produce other NSCs as well as differentiated cells. In order to induce excision of the stop cassette and expression of mutant *Idh1*, NSC lines isolated from embryonic day 14.5 mice were transduced with adenoviral-Cre-recombinase (ad-Cre) or adenoviral-GFP (ad-GFP) as control. When mutant *Idh1* was expressed, the NSCs had a reduced ability to undergo neuronal differentiation and reduced proliferation because of p53-mediated cell cycle arrest. It was also noted in vivo that *Idh1* R132H expression reduced proliferation of cells within the germinal zone of the SVZ [63]. The authors interpreted the results to suggest that mutant IDH1 is detrimental to the glioma cell of origin and the microenvironment.

In order to better understand the mutations involved in gliomagenesis of low-grade glioma, Modrek et al. modeled mutant-IDH1 low-grade glioma formation in NSCs derived from human embryonic stem cells [61]. They sought to investigate glioma progression by introducing the core genetic changes found in low-grade glioma. This consisted of lentiviral expression of R132H-mutant IDH1 and short hairpin RNA (shRNA)-mediated knockdown of p53 and ATRX. Loss of ATRX as the second hit resulted in nonviable cells. They focused on conditions that were most biologically relevant: vector only, mutant IDH1 alone (“1-hit”), mutant IDH1 with p53 knockdown (“2-hit”), and mutant IDH1 with P53 and ATRX knockdown (“3-hit”). Their data supported that gliomagenesis occurs in the order of IDH mutation, then p53 loss, and finally ATRX loss.

To characterize their NSCs as proper models of low-grade glioma, the same researchers profiled NSCs’ DNA methylomes and transcriptomes. All conditions with mutant IDH had elevated levels of global methylation, and when the DNA methylation data of the NSCs with mutant IDH were compared with human low-grade glioma from TCGA [65], there was clustering of IDH1-mutant gliomas with IDH-mutant NSCs. In an analysis of transcriptomic data from RNA-seq, the various IDH-mutant NSCs also clustered with different groups of IDH-mutant low-grade glioma patients from TCGA data sets. Karyotypic analysis revealed that the 3-hit NSCs had significantly elevated numbers of chromosomal fragments, consistent with the genomic instability seen in low-grade gliomas [65,66]. Altogether, these IDH-mutant NSCs had DNA methylome, transcriptome, and karyotype similar to those of low-grade gliomas.

The researchers next found that combined IDH mutation and p53/ATRX loss blocked NSC differentiation. On flow cytometric analysis, 1-hit and 3-hit NSCs had low levels of NSC surface marker CD133 and high levels of restricted glial progenitor marker CD44 compared with vector-only and 2-hit NSCs. Similarly, 1-hit and 3-hit NSCs had near-complete differentiation block when they were directed to differentiate to neurons and astrocytes, while control vector and 2-hit NSCs were able to differentiate. The researchers also identified transcriptional downregulation of SOX2 as a central mechanism underlying the differentiation block [61].

In order to understand the impact of *Idh1* R132H in the context of *Atrx* and *Trp53* loss, Nunez et al. generated an *Idh1*-mutant mouse glioma model using the Sleeping Beauty transposase system [64]. The Sleeping Beauty (SB) transposon system is a nonviral DNA transfer tool that takes advantage of transposable elements (TEs), which are DNA sequences with the ability to move from one genomic location to another. SB takes advantage of the nonreplicative cut-and-paste mechanism that TEs employ in nature. TEs are comprised of a gene encoding the transposase, which is the enzyme that catalyzes the transposition reaction flanked by transposon-specific terminal inverted repeat (TIR) sequences containing binding sites for the transposase. The transposase excises the TE by binding to sequences at the TIRs and inducing double-stranded breaks (DSBs) at both ends. The excised TE then integrates into a different location when the transposase finds a suitable target site and performs the reinsertion of its own genetic code [67]. SB takes advantage of this mechanism by having an expression cassette for the SB transposase and an artificial TE (gene of interest flanked by TIRs) sitting on two separate plasmids such that the transposase is able to stably integrate the gene of interest into the cell’s genome, enabling sustained transgene expression [68].

Nunez et al. used the combination of sh*Trp53*, sh*ATRX*, and mutant *Idh1* with RTK/RAS/PI3K activation to induce gliomas. Increased DNA damage response (DDR) from epigenetic reprogramming of the tumor cells’ transcriptome mediated genomic stability in their IDH1-mutant glioma model in the context of *Atrx* and *p53* knockdown. Furthermore, the researchers found that IDH1 R132H induced transcriptional activation of ataxia-telangiectasia mutated (*Atm),* which resulted in efficient DNA repair activity through homologous recombination DNA repair. This was a clinically relevant finding, as radiation therapy failed to prolong survival in the IDH1-mutant tumor-bearing mice, but pharmacological inhibition of DDR prolonged survival due to radiosensitivity. Taken together, these findings opened up the potential that DDR inhibition combined with radiation therapy could be a novel therapeutic approach for IDH1 R132H glioma patients harboring *ATRX* and *TP53* inactivating mutations.

### 5.2. Immune Microenvironment

Some in vivo models have shed light on the unique role of the tumor microenvironment in the development of low-grade glioma. To study the progressive change in the actions of immune cells and glioma cells, Appolloni et al. used a glioma mouse model driven by the overexpression of the *Pdgfb* oncogene [59]. In their model, multifocal gliomas were first generated by injecting E14 embryos with replication-incompetent retroviral vectors expressing PDGFB in the lateral ventricles. Early-onset tumors displayed histological features of low-grade tumors, while late-onset tumors showed those of high-grade tumors. Orthotopic transplantation of early-onset gliomas did not produce secondary tumors, while injection of late-onset tumors did. However, injection of early-onset tumors in NOD/SCID immunocompromised mice did result in tumors demonstrating a role of the adaptive immune system. When secondary tumors from early-onset tumors were injected in NOD/SCID immunocompetent mice, they were able to generate tumors. The authors interpreted this to indicate that the residual immune system components in NOD/SCID mice enabled tumors to progress toward higher grades. Genetic analysis showed a downregulation of immune response and inflammation genes in the late-onset tumors compared to the early-onset tumors. Furthermore, they found greater infiltration by CD8-positive lymphocytes in the brains of mice with low-grade tumors, suggesting that low-grade, but not high-grade, gliomas stimulate such infiltration. Overall, their work showed the important role of the immune system on patterns of progression to malignancy over time.

Further work on mutant IDH focused on investigating the effect of this mutation on the immunologic tumor microenvironment. Amankulor et al. employed a strategy using mutant IDH1 and wildtype IDH1 mouse glioma models whose initiating events were identical, with the exception of *Idh1* mutation status. This was conducted using the RCAS/TVA system to ectopically express mutant IDH1 (R132H) in PDGF-driven gliomas [60]. In this system, RCAS retroviral vectors transfer genes into cells that express the Tva receptor. Three different mouse strains were used in which the Tva receptor was expressed from the *Nestin* promoter (Ntva): Ntva_*Ink4a*/*Arf*^−/−^, Ntva_*Ink4a*/*Arf*^+/−^, and Ntva_*Ink4a*/*Arf*^+/+^. In this fashion, RCAS would transfer genes to CNS progenitors. RCAS–PDGF-producing DF1 cells were injected with either DF1 cells producing RCAS–wildtype *Idh1*-sh*Trp53* (wtIdh1) or RCAS–mutant Idh1-sh*Trp53* (muIdh1) in mice. Thus, the researchers were able to generate tumors that had the same genetic background and differed only in *Idh1* mutation status. When the RNA expression patterns were compared between mutant *Idh1* mouse glioma and wildtype *Idh1* mouse glioma, a differential association with gene expression of immune system processes was found, with wildtype *Idh1* mouse gliomas having strong associations with the positive regulation of immune responses. Genes related to leukocyte and neutrophil migration were relatively downregulated in muIdh1 mouse gliomas. Furthermore, flow cytometric analysis of single-cell suspensions from brain tissue of mice with IDH1-mutant gliomas displayed significantly fewer CD45+ immune cells compared with brain tissue of mice with wildtype *Idh1* gliomas. To investigate the extent of neutrophil chemotaxis in mutant IDH1 gliomas, the researchers conducted migration experiments on neutrophils in the presence of muIdh1 cells. Their data indicated that chemotaxis to muIdh1 gliomas was repressed, as tissue homogenates of wtIdh1 mouse gliomas had roughly double the migration index of muIdh1 mouse gliomas. Conditioned medium experiments further suggested that tissue homogenates from IDH1-mutant mouse gliomas may contain lower levels of neutrophil chemoattractants. Genetic and proteomic analysis of muIdh1 mouse glioma tissues demonstrated the downregulation of cytokine protein expression.

In the same study, the researchers assessed whether differences in the immune microenvironment were the biological cause of survival differences between mice with muIdh1 versus wtIdh1 gliomas. Intriguingly, treatment of wtIdh1 and muIdh1 tumor-bearing mice with an anti-Ly6g (1A8) or isotype control (2A3) antibody to deplete neutrophil populations demonstrated a significant survival benefit for mice with wtIdh1 tumors with neutrophil depletion, but no significant effect was seen on mice with muIdh1 tumors. Taken together, these findings demonstrated that muIdh1 has unique effects on the immune microenvironment, which may have a role in the survival differences between muIdh1 and wtIdh1 tumors.

Liu et al. went further in trying to understand the role of mutant IDH1 in gliomagenesis using a preclinical model of low-grade glioma. They used two samples of freshly resected low-grade gliomas, with one being an astrocytoma with mutant IDH1 R132C and another being an oligodendroglioma with mutant IDH1 R132H [62]. They generated induced pluripotent stem cells (iPSCs) from these lines by transducing them with the reprogramming factors *OCT4*, *SOX2*, *KLF4*, and *c-MYC*. The resultant low-grade glioma iPSCs (LGG-iPSCs) were confirmed to be reprogrammed, as they had pluripotency markers and were capable of differentiating into tissues from all three embryonic germ layers. Intriguingly, these reprogrammed cells no longer contained *IDH1* mutations, indicating that IDH1 mutations inhibit somatic reprogramming. Array-based comparative genomic hybridization analysis on primary low-grade glioma cells and the derived LGG-iPSCs demonstrated regional amplifications on chromosome Xq23 on astrocytoma LGG-iPSCs and 7q31 on oligodendroglioma LGG-iPSCs. These regional amplifications were not seen in the genomes of healthy individuals but were seen in higher frequency in low-grade gliomas [62]. The researchers reasoned that these regional amplifications are early genetic lesions that occur before *IDH1* mutations and that the mutation in *IDH1* is likely not the initiating factor in gliomagenesis.

## 6. Conclusions

Continuous progress is being made on the important endeavor of creating ideal preclinical models of low-grade glioma. This effort has been more challenging than creating models of high-grade gliomas, which often have greater success rates. However, adequate models are crucial for the advancement of our understanding of low-grade glioma. The attempts to circumvent many of the challenges in developing these models have shed light on our understanding of the fundamental biology that drives gliomagenesis. Those studies have also highlighted the importance of factors beyond genetics, such as the immune microenvironment of the tumor. Further research is needed to yield models that will more closely parallel human gliomas, creating the ideal system to develop and test novel therapeutics for the future.

## Figures and Tables

**Table 1 cancers-15-00596-t001:** Vectors used for gene delivery in LGG models [58].

Viral or Nonviral	Vector Type	Advantage	Limitations	Example
Viral	Retrovirus	Cell-specific infection	Safety concerns when transducing oncogenes	[48,59,60]
Viral	Lentivirus	Infects both dividing and nondividing cells	Safety concerns when transducing oncogenes	[61,62]
Viral	Adenovirus	No genome integration	High immunogenicity	[63]
Nonviral	Nonviral transposon (Sleeping Beauty)	Suitable for discovery of tumor drivers	Genome integration may disrupt gene expression	[64]

**Table 2 cancers-15-00596-t002:** LGG models.

Model Category	Specific Model	Genes Involved	Genetic Heterogeneity	Immunocompetent	Brain Micro-Environment	Blood–Brain Barrier	Reproducible
Murine cell line	ENU-induced murine tumor cells	BRAF, TP53, PDGFRa, CDKN2a, EGFR, and no IDH	yes	no	no	no	no
Murine cell line	IDH1 mutant expression in SVC cells	IDH	no	no	no	no	yes
Patient-derived cell line	BT142	IDH	no	no	no	no	yes
Patient-derived cell line	TS603	IDH, 1p/19q codeletion	no	no	no	no	yes
Murine-derived GEMMs	Sleeping Beauty transposase system	IDH, TP53, and ATRX	no	yes	yes	yes	yes
Murine-derived GEMMs	RCAS-mutIDH-PDGFA-CDKN2A-ATRX-PTEN	IDH, CDKN2a, ATRX, PTEN, and PDGFA	no	yes	yes	yes	yes
Murine derived GEMMs	RCAS-mutIDH-PDGF driven-p53 knockdown	IDH, PDGF, and TP53	no	yes	yes	yes	yes
Patient-derived murine model	Various LGG orthotopic xenografts	IDH, FUBP1, and CIC	partially	no	partially	yes	yes
Genetically modified neurosphere	hESCs with lentiviral modification	IDH, TP53, and ATRX	no	no	no	no	yes
Mouse to mouse xenograft	PDGF-B overexpressing mouse NSC into mouse brain	PDGF-B	partially	yes	yes	yes	yes
iPSC	human LGG iPSC	IDH	partially	no	no	no	yes

## Data Availability

Data sharing is not applicable to this article.

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
