# Peer review of "Preclinical Models of Low-Grade Gliomas"

_cancers, 2023, doi:10.3390/cancers15030596_

Round 1

Reviewer 1 Report

This is a concise overview of the topic. Pre-clinical models of brain cancer in general have shown poor fidelity for clinical translation especially in the field of drug development. This is particularly acute in LGG

I think the review would be strengthened by a section on patient avatars of low-grade gliomas and also the potential of in silico models

Author Response

Thanks a section on patient avatars of low-grade gliomas and also a section on the potential of in silico models have been added. 

Reviewer 2 Report

This review article describes the current low-grade glioma experimental models and the challenges associated with developing and maintaining such models. 

The review is quite comprehensive and well-referenced although further elaboration on important systems (such as Sleeping Beauty) would increase the value of this review, especially for less experienced scientists. A more detailed description of the vectors used for gene delivery in mouse glioma models, possibly with tabular formats for added clarity would also be helpful. A guiding reference for both of these suggestions follows: 

https://doi.org/10.1007/978-1-0716-0856-2_1

Author Response

Thanks, further elaboration on important systems such as Sleeping Beauty was added into the manuscript and a table further describing the vectors used for gene delivery in LGG models was also incorporated.